# Marginalized Importance Sampling for Off-Environment Policy Evaluation

**Pulkit Katdare**
Department of Electrical and Computer Engineering
University of Illinois Urbana-Champaign Illinois, United States
`katdare2@illinois.edu`

**Nan Jiang**
Department of Computer Science
University of Illinois Urbana-Champaign Illinois, United States

**Katherine Driggs-Campbell**
Department of Electrical and Computer Engineering
University of Illinois Urbana-Champaign Illinois, United States

**Abstract:** Reinforcement Learning (RL) methods are typically sample-inefficient, making it challenging to train and deploy RL-policies in real world robots. Even a robust policy trained in simulation requires a real-world deployment to assess their performance. This paper proposes a new approach to evaluate the real-world performance of agent policies prior to deploying them in the real world. Our approach incorporates a simulator along with real-world offline data to evaluate the performance of any policy using the framework of Marginalized Importance Sampling (MIS). Existing MIS methods face two challenges: (1) large density ratios that deviate from a reasonable range and (2) indirect supervision, where the ratio needs to be inferred indirectly, thus exacerbating estimation error. Our approach addresses these challenges by introducing the target policy's occupancy in the simulator as an intermediate variable and learning the density ratio as the product of two terms that can be learned separately. The first term is learned with direct supervision and the second term has a small magnitude, thus making it computationally efficient. We analyze the sample complexity as well as error propagation of our two step-procedure. Furthermore, we empirically evaluate our approach on Sim2Sim environments such as Cartpole, Reacher, and Half-Cheetah. Our results show that our method generalizes well across a variety of Sim2Sim gap, target policies and offline data collection policies. We also demonstrate the performance of our algorithm on a Sim2Real task of validating the performance of a 7 DoF robotic arm using offline data along with the Gazebo simulator.

**Keywords:** Sim2Real, Policy Evaluation, Robot Validation

## 1 Introduction

Reinforcement Learning (RL) algorithms aim to select actions that maximize the cumulative returns over a finite time horizon. In recent years, RL has shown state-of-the-art performance over a range of complex tasks such as chatbots, [1], games [2], and robotics [3, 4, 5, 6]. However, RL algorithms still require a large number of samples, which can limit their practical use in robotics [7, 8].

A typical approach is to train robust robot policies in simulation and then deploy them on the robot [9, 10]. Such an approach, although useful, does not guarantee optimal performance on the real robot without significant fine tuning [11]. In this work, we propose an approach that evaluates

7th Conference on Robot Learning (CoRL 2023), Atlanta, USA.

the real world performance of a policy using a robot simulator and offline data collected from the real-world [6]. To achieve this, we employ the framework of off-policy evaluation (OPE) [12].

OPE is the problem of using offline data collected from a possibly unknown *behavior* policy to estimate the performance of a different *target* policy. Classical OPE methods are based on the principle of importance sampling (IS) [13, 14, 15], which reweights each trajectory by its density ratio under the target versus the behavior policies. More recently, significant progress has been made on *marginalized importance sampling* (MIS), which reweights transition tuples using the density ratio (or importance weights) over states instead of trajectories to overcome the so-called curse of horizon [16, 17, 18]. The density ratio is often learned via optimizing minimax loss functions.

Most existing MIS methods are model-free, relying on data from the real environment to approximate the MIS weight function. However, a direct application of MIS methods to robotics carries two main disadvantages. (1) *Large ratios:* MIS method learns distribution mismatch between the behavior and the target policies. When the mismatch between the behavior and the target policy is large, which is often the case, MIS method tend be challenging to learn. (2) *Indirect supervision:* MIS methods requires no samples from target policies, which requires the weight being learned *indirectly* via the Bellman flow equation. In states where coverage of the offline dataset is scarce, MIS methods tend to perform poorly.

In robotics, it is reasonable to assume access to a good but imperfect simulator of the real environment [19, 20, 21]. In this work, we propose an improved MIS estimator that estimates the density ratio mismatch between the real world and the simulator. We further show that such a MIS estimator can be used to evaluate the real-world performance of a robot using just the simulator. As described in figure 1, we estimate the discrepancy between the real world and the simulator by using the target policy's occupancy in the simulator as an intermediate variable. This allows us to calculate the MIS weights as a combination of two factors, which can be learned independently. The first factor has direct supervision, while the second factor has a small magnitude (close to 1), thereby addressing both large ratios and indirect supervision issues mentioned above. We present a straightforward derivation of our method, examine its theoretical properties, and compare it to existing methods (including existing ways of incorporating a simulator in OPE) and baselines through empirical analysis.

We make the following contributions. (1) We derive an MIS estimator for off-environment evaluation (Section 4). (2) We explore the theoretical properties of our off-environment evaluation estimator by proposing a sample-complexity analysis (Section 4) and studying its special cases in linear and kernel settings. (3) We empirically evaluate our estimator on both Sim2Sim as well as Sim2Real environments (Section 5). For the Sim2Sim experiments, we perform a thorough ablation study over different Sim2Sim gap, data-collection policies and target policies, environments (Taxi, Cartpole, Reacher, and Half-Cheetah). Furthermore, we demonstrate practicality of our approach on a sim2real task by validating performance of a Kinova robotic arm over using offline data along with Gazebo based Kinova simulator.

## 2   Preliminaries

Robot learning problems are often modelled as an infinite-horizon discounted Markov Decision Process (MDP). MDP is specified by $(\mathcal{S}, \mathcal{A}, P, R, \gamma, d_0)$. Here, $\mathcal{S}$ and $\mathcal{A}$ are the state and the action spaces, $P : \mathcal{S} \times \mathcal{A} \to \Delta(\mathcal{S})$ is the transition function ($\Delta(\cdot)$ is the probability simplex). We also define the reward function $R : \mathcal{S} \times \mathcal{A} \to \Delta([0, R_{max}])$, $\gamma \in [0, 1)$ is the discount factor, and $d_0 \in \Delta(\mathcal{S})$ is the initial state distribution. A policy $\pi : \mathcal{S} \to \Delta(\mathcal{A})$ induces a distribution of trajectory: $\forall t \geq 0, s_0 \sim d_0, a_t \sim \pi(\cdot|s_t), r_t \sim R(\cdot|s, a), s_{t+1} \sim P(\cdot|s_t, a_t)$. The performance of $\pi$ is measured by its expected discounted return under the initial state distribution, defined as $J_P(\pi) = (1 - \gamma)\mathbb{E}[\sum_{t=0}^{\infty} \gamma^t r_t | \pi, d_0]$; here, we use the subscript $P$ in $J_P(\pi)$ to emphasize the dynamics w.r.t. which the return is defined, since we will consider both the true environment and the simulator in the rest of this paper and the subscript will help distinguish between them. $J_P(\pi)$ also

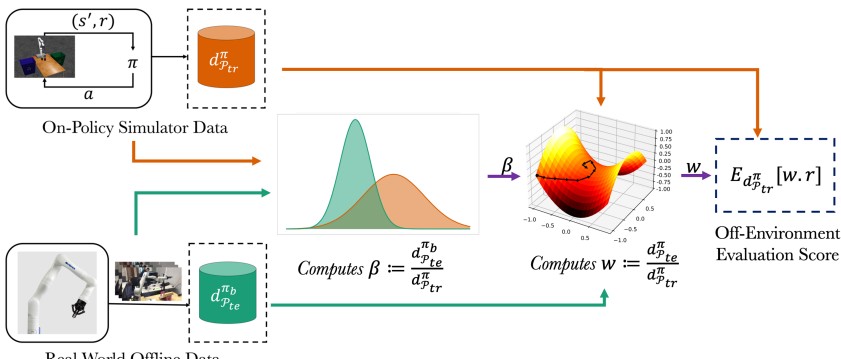

Figure 1: For a given policy $\pi$, we first collect on-policy data $d^\pi_{\mathcal{P}_{tr}}$ from a simulator environment. Using $d^\pi_{\mathcal{P}_{tr}}$ and offline data $d^{\mathcal{D}}_{\mathcal{P}_{te}}$ on the real world, we first calculate the importance sampling factor $\beta = d^{\pi_b}_{\mathcal{P}_{te}}/d^\pi_{\mathcal{P}_{tr}}$. This importance sampling factor essentially allows us to re-weight existing off-policy evaluation algorithm to estimate $w = d^\pi_{\mathcal{P}_{te}}/d^\pi_{\mathcal{P}_{tr}}$ which helps us estimate real-world performance of the agent $J_{\mathcal{P}_{te}}(\pi)$ using on-policy simulator data.

has an alternative expression $J_P(\pi) := \mathbb{E}_{(s,a)\sim d^\pi_P, r\sim R(s,a)}[r]$, where

$$d^\pi_P(s,a) = (1-\gamma)\sum_{t=0}^\infty \gamma^t \mathbb{P}[s_t = s, a_t = a | \pi, d_0] \tag{1}$$

is the discounted state-action occupancy induced by $\pi$ from $d_0$. An important quantity associated with a policy is its Q-function $Q^\pi_P$, which satisfies the Bellman equation $Q^\pi_P(s,a) = \mathbb{E}_{r\sim R(s,a), s'\sim P(s,a)}[r + \gamma Q^\pi_P(s', \pi)]$, where $f(s', \pi) := \mathbb{E}_{a'\sim\pi(\cdot|s')}[f(s', a')]$. We can also define the state-value function $V^\pi_P(s) = Q^\pi_P(s, \pi)$, and $J(\pi) = (1-\gamma)\mathbb{E}_{s\sim d_0}[V^\pi_P(s)]$.

**OPE and Marginalized Importance Sampling:** In off-policy evaluation (OPE), we want to evaluate a target policy $\pi$ using data collected from a different policy in the real environment, denoted by its dynamics $P$. As a standard simplification, we assume data is generated i.i.d. as $(s,a)\sim\mu, r\sim R(s,a), s'\sim P(s,a)$, and the sample size is $n$. When the data is generated from some behavior policy $\pi_b$, $\mu$ can correspond to its occupancy $d^{\pi_b}_P$. We will use $\mathbb{E}_\mu[\cdot]$ as a shorthand for taking expectation over $(s,a,r,s')$ generated from such a data distribution in the real environment.

The key idea in marginalized importance sampling (MIS) is to learn the weight function $w^{\pi/\mu}_P(s,a) := \frac{d^\pi_P(s,a)}{\mu(s,a)}$. Once this function is known, $J(\pi)$ can be estimated as $J(\pi) = \mathbb{E}_{(s,a)\sim d^\pi_P, r\sim R(s,a)}[r] = \mathbb{E}_\mu[w^{\pi/\mu}_P(s,a)\cdot r]$. Note that $\mathbb{E}_\mu[\cdot]$ can be empirically approximated by the dataset sampled from the real environment. The real challenge in MIS is how to learn $w^{\pi/\mu}_P$. Existing works often do so by optimizing minimax loss functions using Q-functions as discriminators, and is subject to both difficulties (large ratios and indirect supervision) mentioned in the introduction. We refer the readers to [22] for a summary of typical MIS methods.

**Learning Density Ratios from Direct Supervision:** Given two distributions $p$ and $q$ over the same space $\mathcal{X}$, the density ratio $p(x)/q(x)$ can be learned *directly* if we have access to samples from both $p$ and $q$, using the method proposed by [23]:

$$\frac{p(x)}{q(x)} = \arg\max_{f:\mathcal{X}\to\mathbb{R}_{>0}} \mathbb{E}_{x\sim p}[\ln f(x)] - \mathbb{E}_{x\sim q}[f(x)] + 1. \tag{2}$$

To guarantee generalization over a finite sample when we approximate the expectations empirically, we will need to restrict the space of $f$ that we search over to function classes of limited capacities, such as RKHS or neural nets, and $p(x)/q(x)$ can still be well approximated as long as it can be represented in the chosen function class (i.e., *realizable*). More concretely, if we have $n$ samples $x_1, \ldots, x_n$ from $p$ and $m$ samples $\tilde{x}_1, \ldots, \tilde{x}_m$ from $q$, and use $\mathcal{F}$ to approximate $p(x)/q(x)$, the learned density ratio can be made generalizable by adding a regularization term $I(f)$ to improve the statistical and computational stability of learning:

$$\arg\max_{f\in\mathcal{F}} \frac{1}{n}\sum_i \ln f(x_i) - \frac{1}{m}\sum_j f(\tilde{x}_j) + \frac{\lambda}{2}I(f)^2. \tag{3}$$

# 3 Related Work

## 3.1 Reinforcement Learning applications in Robotics

There are three different themes that arise in reinforcement learning for robotics [24, 25]. (1) **Sim2Real:** algorithms are primarily concerned with learning robust policy in simulation by training the algorithms over a variety of simulation configurations [26, 9, 27]. Sim2Real algorithms, although successful, still require a thorough real-world deployment in order to gauge the policy's performance. (2) **Imitation learning** algorithms learn an optimal policy by trying to mimic offline expert demonstrations [28, 29, 30]. Many successful imitation learning algorithms minimize some form of density matching between the expert demonstrations and on-policy data to learn optimal policy. A key problem with imitation learning is the fact that it requires constant interaction with the real-world environment in order to learn an optimal policy. (3) **Offline reinforcement learning** is a relatively new area. Here the idea is to learn an optimal policy using offline data without any interaction with the environment [31, 32, 33, 34]. Offline reinforcement learning has recently demonstrated performance at-par with classical reinforcement learning in a few tasks. However, offline reinforcement learning algorithms tend to overfit on the offline data. Thus, even offline learning modules too require an actual deployment in-order to assess performance.

## 3.2 Off-Policy Evaluation

We review related works in this section, focusing on comparing to existing OPE methods that can leverage the side information provided by an imperfect simulator. (1) **Marginalized Importance Sampling (MIS):** MIS methods tend to assume the framework of data collection and policy evaluation being on the same environment. To that end, there are both model-free [35, 36, 37] and model-based variants of MIS methods and face the aforementioned two challenges (large magnitude of weights and indirection supervision) simultaneously. Model based variants of MIS sometimes tend to be doubly robust (DR) in nature [18, 38] and can benefit from Q-functions as control variates, which can be supplied by the simulator. However, the DR version of MIS is a meta estimator, and the weight $d^\pi_{P_{te}}/\mu$ still needs to be estimated via a "base" MIS procedure. Therefore, the incorporation of the simulator information does not directly address the challenges we are concerned with, and there is also opportunity to further combine our estimator into the DR form of MIS. (2) **Model-based methods:** Model-based estimators first approximate the transition dynamics of the target environment [39], which is further used to generate rollouts and evaluate performance for any target policy. One way of incorporating a given imperfect simulator in this approach is to use the simulator as "base predictions," and only learn an additive correction term, often known as residual dynamics [40]. This approach combines the two sources of information (simulator and data from target environment) in a very different way compared to ours, and are more vulnerable to misspecification errors than model-free methods.

# 4 Weight Estimator

Recall that our goal is to incorporate a given simulator into MIS. We will assume that the simulator shares the same $\mathcal{S}, \mathcal{A}, \gamma, d_0$ with the real environment, but has its own transition function $P_{tr}$ which can be different from $P_{te}$. As we will see, extension to the case where the reward function is unknown and must be inferred from sample rewards in the data is straightforward, and for simplicity we will only consider difference in dynamics for most of the paper.

**Split the weight:** The key idea in our approach is to split the weight $w_P^{\pi/\mu}$ into two parts by introducing $d^\pi_{P_{tr}}$ as an intermediate variable:

$$w_P^{\pi/\mu}(s,a) = \frac{d^\pi_{P_{te}}(s,a)}{\mu(s,a)} = \underbrace{\frac{d^\pi_{P_{tr}}(s,a)}{\mu(s,a)}}_{\substack{:=\beta \\ \text{(direct supervision)}}} \cdot \underbrace{\frac{d^\pi_{P_{te}}(s,a)}{d^\pi_{P_{tr}}(s,a)}}_{:=w^\pi_{P_{te}/P_{tr}} \text{(magnitude} \simeq 1)}$$

Note that $d^\pi_{P_{tr}}$ is the occupancy of $\pi$ in the simulator, which we have free access to. The advantage of our approach is that by estimating $\beta$ and $w^\pi_{P_{te}/P_{tr}}$ separately, we avoid the situation of running

into the two challenges mentioned before simultaneously, and instead address one at each time: $\beta = d_{P_{tr}}^{\pi}/\mu$ has large magnitude but can be learned directly via [23] without the difficult minimax optimization typically required by MIS, and we expect $w_{P_{te}/P_{tr}}^{\pi} = d_{P_{te}}^{\pi}/d_{P_{tr}}^{\pi}$ to be close to 1 when $P_{te} \approx P_{tr}$ (and thus easier to learn).

**Estimate $w_{P_{te}/P_{tr}}^{\pi}$:** Since $\beta$ is handled by the method of [23], the key remaining challenge is how to estimate $w_{P_{te}/P_{tr}}^{\pi}$. (Interestingly, $\beta$ also plays a key role in estimating $w_{P_{te}/P_{tr}}^{\pi}$, as will be shown below.) Note that once we have approximated $w_{P_{te}/P_{tr}}^{\pi}$ with some $w$, we can directly reweight the state-action pairs from the simulator (i.e., $d_{P_{tr}}^{\pi}$) if the reward function is known (this is only assumed for the purpose of derivation), i.e.,

$$J_{P_{te}}(\pi) \approx \mathbb{E}_{(s,a)\sim d_{P_{tr}}^{\pi}, r\sim R(s,a)}[w \cdot r],$$

and this becomes an identity if $w = w_{P_{te}/P_{tr}}^{\pi}$. Following the derivation in [18, 22], we now reason about the error of the above estimator for an arbitrary $w$ to derive an upper bound as our loss for learning $w$:

$$|\mathbb{E}_{(s,a)\sim d_{P_{tr}}^{\pi}, r\sim R(s,a)}[w \cdot r] - J_{P_{te}}(\pi)|$$

$$= |\mathbb{E}_{(s,a)\sim d_{P_{tr}}^{\pi}, s'\sim P(s,a)}[w \cdot (Q_{P_{te}}^{\pi}(s,a) - \gamma Q_{P_{te}}^{\pi}(s',\pi))] - (1-\gamma)\mathbb{E}_{s\sim d_0}[Q_{P_{te}}^{\pi}(s,\pi)]|$$

$$\leq \sup_{q\in\mathcal{Q}} |\mathbb{E}_{d_{P_{tr}}^{\pi}\times P_{te}}[w \cdot (q(s,a) - \gamma q(s',\pi))] - (1-\gamma)\mathbb{E}_{s\sim d_0}[q(s,\pi)]|. \tag{4}$$

Here $d_{P_{tr}}^{\pi} \times P_{te}$ is a shorthand for $(s,a) \sim d_{P_{tr}}^{\pi}, s' \sim P(s,a)$. In the last step, we handle the unknown $Q_P^{\pi}$ by a relaxation similar to [16, 18, 22], which takes an upper bound of the error over $q \in \mathcal{Q}$ for some function class $\mathcal{Q} \subset \mathbb{R}^{\mathcal{S}\times\mathcal{A}}$, and the inequality holds as long as $Q_P^{\pi} \in conv(\mathcal{Q})$ with $conv(\cdot)$ being the convex hull.

**Approximate $d_{P_{tr}}^{\pi} \times P_{te}$:** The remaining difficulty is that we will need samples from $d_{P_{tr}}^{\pi} \times P_{te}$, i.e., $(s,a)$ sampled from $\pi$'s occupancy in the $P_{tr}$ *simulator*, and the next $s'$ generated in the $P_{te}$ *real environment*. While there is no natural dataset for such a distribution, we can take the data from the real environment, $(s,a,s') \sim \mu \times P_{te}$, and reweight it using $\beta = d_{P_{tr}}^{\pi}/\mu$ to approximate expectation w.r.t. $d_{P_{tr}}^{\pi} \times P_{te}$, i.e.,

$$(s,a,s') \sim d_{P_{tr}}^{\pi} \times P_{te} \quad \Longleftrightarrow \quad (s,a,s') \sim \mu \times P_{te} \text{ reweighted with } \beta := d_{P_{tr}}^{\pi}/\mu.$$

Based on such an observation, we can further upper-bound $|\mathbb{E}_{(s,a)\sim d_{P_{tr}}^{\pi}, r\sim R(s,a)}[w \cdot r] - J_{P_{te}}(\pi)|$ from end of Equation 4 with:

$$\sup_{q\in\mathcal{Q}} L_w(w,\beta,q) := |\mathbb{E}_\mu[w \cdot \beta \cdot (q(s,a) - \gamma q(s',\pi))] - (1-\gamma)\mathbb{E}_{s\sim d_0}[q(s,\pi)]|. \tag{5}$$

As our derivation has shown, this is a valid upper bound of the error as long as $conv(\mathcal{Q})$ can represent $Q_{P_{te}}^{\pi}$. We also need to show that the upper bound is non-trivial, i.e., when $w = w_{P_{te}/P_{tr}}^{\pi}$, the upper bound should be 0. This is actually easy to see, as for any $q$:

$$L(w_{P_{te}/P_{tr}}^{\pi}, \beta, q) := |\mathbb{E}_{d_{P_{tr}}^{\pi}\times P_{te}}[w_{P_{te}/P_{tr}}^{\pi} \cdot (q(s,a) - \gamma q(s',\pi))] - (1-\gamma)\mathbb{E}_{s\sim d_0}[q(s,\pi)]| \tag{6}$$

$$= |\mathbb{E}_{d_{P_{te}}^{\pi}\times P}[q(s,a) - \gamma q(s',\pi)] - (1-\gamma)\mathbb{E}_{s\sim d_0}[q(s,\pi)]| = 0. \tag{7}$$

The last step directly follows from the fact that $d_{P_{te}}^{\pi}$ is a valid discounted occupancy and obeys the Bellman flow equation. Therefore, it makes sense to search for $w$ over a function class $\mathcal{W} \subset \mathbb{R}^{\mathcal{S}\times\mathcal{A}}$ to minimize the loss $\sup_{q\in\mathcal{Q}} L(w,\beta,q)$.

**Final estimator:** To summarize our estimation procedure, we will first use [23] to estimate $\hat{\beta} \approx d_{P'}^{\pi}/\mu$ with a function class $\mathcal{F}$, and plug the solution into our loss for estimating $w_{P_{te}/P_{tr}}^{\pi}$, i.e.,

$$\hat{w} = \arg\min_{w\in\mathcal{W}} \sup_{q\in\mathcal{Q}} L_w(w,\hat{\beta},q). \tag{8}$$

As mentioned above, if the reward function is known, we can use $\mathbb{E}_{(s,a)\sim d_{P_{tr}}^{\pi}, r\sim R(s,a)}[\hat{w} \cdot r]$ as our estimation of $J_{P_{te}}(\pi)$. We can also demonstrate interesting properties of our optimization like the effect of a linear function class and RKHS function class. This discussion is deferred to the supplementary materials section 9.1.

**Sample Complexity Guarantee:** We can further provide an upper-bound on the performance of our final estimator under the following two assumptions.

**Assumption 1** (Boundedness). *We assume $\forall f \in \mathcal{F}$, $0 < C_{\mathcal{F},\min} \leq f \leq C_{\mathcal{F},\max}$. Define $C_{\mathcal{F}} := C_{\mathcal{F},\max} + \max(\log C_{\mathcal{F},\max}, -\log C_{\mathcal{F},\min})$. Similarly, $\forall w \in \mathcal{W}$, $w \in [0, C_{\mathcal{W}}]$, and $\forall q \in \mathcal{Q}$, $q \in [0, C_{\mathcal{Q}}]$.*

**Assumption 2** (Realizability of $\mathcal{F}$). *$d_{P'}^{\pi}/\mu \in \mathcal{F}$.*

**Theorem 4.1.** *Let $\hat{\beta}$ be our approximation of $d_{P'}^{\pi}/\mu$ which we found using [23]. We utilize this $\hat{\beta}$ to further optimize for $\hat{w}_n$ (equation 8) using $n$ samples. In both cases, $\mathbb{E}_{(s,a)\sim d_{P_{tr}}^{\pi}}[\cdot]$ is also approximated with $n$ samples from the simulator $P_{tr}$. Then, under Assumptions 1 and 2 along with the additional assumption that $Q_{P_{te}}^{\pi} \in C(\mathcal{Q})$ with probability at least $1 - \delta$, the Off Environment Evaluation error can be bounded as*

$$\left|\mathbb{E}_{(s,a)\sim d_{P_{tr}}^{\pi}, r\sim R(s,a)}[\hat{w}_n \cdot r] - J_{P_{te}}(\pi)\right| \leq \min_{w \in \mathcal{W}} \max_{q \in \mathcal{Q}} |L_w(w, \beta, q)|$$

$$+ 2C_{\mathcal{W}} \cdot C_{\mathcal{Q}} \cdot \tilde{O}\left(\sqrt{\left\|\frac{d_{P'}^{\pi}}{\mu}\right\|_{\infty} \cdot \left(4\mathbb{E}\mathcal{R}_n(\mathcal{F}) + C_{\mathcal{F}}\sqrt{\frac{2\log(\frac{2}{\delta})}{n}}\right)}\right) \tag{9}$$

$$+ 2\mathcal{R}_n(\mathcal{W}, \mathcal{Q}) + C_{\mathcal{Q}}C_{\mathcal{W}}\sqrt{\frac{\log(\frac{2}{\delta})}{2n}}$$

*where $\mathcal{R}_n(\mathcal{F}), \mathcal{R}_n(\mathcal{W}, \mathcal{Q})$ are the Radamacher complexities of function classes $\{(x, y) \to f(x) - \log(f(y)) : f \in \mathcal{F}\}$ and $\{(s, a, s') \to (w(s, a) \cdot \frac{d_{P'}^{\pi}(s,a)}{\mu(s,a)} \cdot (q(s, a) - \gamma q(s', \pi)) : w \in \mathcal{W}, q \in \mathcal{Q}\}$, respectively, $\|d_{P'}^{\pi}/\mu\|_{\infty} := \max_{s,a} d_{P'}^{\pi}(s, a)/\mu(s, a)$ measures the distribution shift between $d_{P'}^{\pi}$ and $\mu$, and $\tilde{O}(\cdot)$ is the big-Oh notation suppressing logarithmic factors.*

Note that we do not make realizability assumption for $\mathcal{W}$ in the theorem above. Realizability assumption is reflected in the $\inf_{w \in \mathcal{W}} \sup_{q \in \mathcal{Q}} |L(w, \beta, q)|$, which equals 0 when $d_{P_{te}}^{\pi}/d_{P_{tr}}^{\pi} \in \mathcal{W}$. The remaining terms vanishes to 0 at an $O(1/\sqrt{n})$ rate when $n \to \infty$.

**Generalizing Off-Environment Policy Evaluation:** A key advantage of our two-step approach is that we can improve many existing off-policy evaluation algorithm with a similar two-step process. In this work, we use our two-step procedure with GradientDICE—which is an empirically state-of-the-art estimator in the DICE family—can also be similarly adapted as below, which we use in our experiments. Detailed derivation for the same can be found in the supplementary materials 9.3.

## 5 Experiments

### 5.1 Sim2Sim Validation of $\beta$-DICE

**Experimental Setting:** In the Sim2Sim experiments, we aim to show the effectiveness of our approach across different target policies, offline dataset as well as changing sim2sim gap. We further show the effectiveness of our approach over different types of Sim2Sim environments like Taxi (Tabular), Cartpole (discrete-control), Reacher (continuous control), and HalfCheetah (continuous control) environments. For each of these environments, we refer to the default configurations of

Table 1: Log mean squared error between the performance predicted by our $\beta$-DICE algorithm and the real world performance of the robot. We observe that our method is able to outperform DICE based baselines by a comfortable margin.

| Algorithm | $\mathbf{Log_{10}}$ **Mean Squared Error** ($\downarrow$) | | | | |
| --- | --- | --- | --- | --- | --- |
| | Kinova (Sim2Real) | Taxi | Cartpole | Reacher | Half-Cheetah |
| Simulator | -3.96 | -0.19 | -2.58 | -1.09 | 1.18 |
| $\beta$-DICE (Ours) | **-4.38** | **-1.60** | **-4.19** | **-4.08** | **-3.42** |
| GenDICE | -3.48 | -0.13 | -2.84 | -2.61 | -2.96 |
| GradientDICE | -3.49 | -0.59 | -1.45 | -3.17 | -2.16 |
| DualDICE | -3.48 | -0.48 | -0.99 | -2.88 | -2.12 |

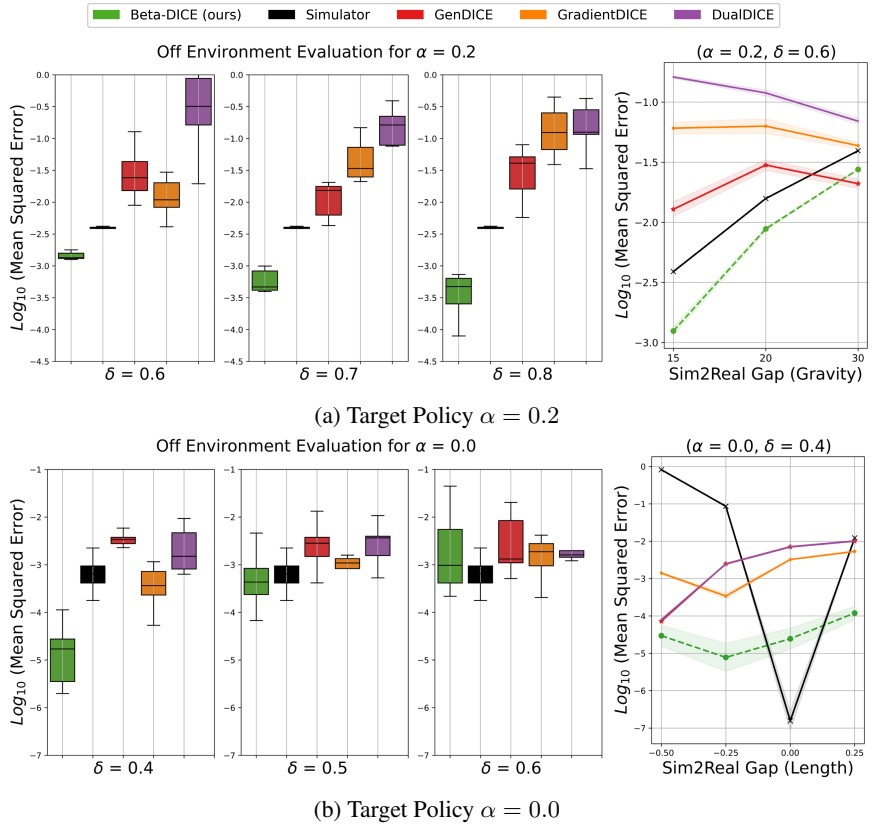

Figure 2: We demonstrate the effectiveness of $\beta$-DICE on Cartpole (a) and Reacher (b) Sim2Sim environment. For Cartpole environment we demonstrate the performance of $\beta$-DICE over for a Sim2Sim pair of $\{10, 15\}m/s^2$. Similarly for Reacher the Sim2Sim pair is (0.1m, 0.075m) for the length of the link. On left hand side, we demonstrate the effect of $\beta$-DICE with different data collection policies while keeping the target policies fixed. On the right hand side, we demonstrate the impact of $\beta$-DICE with increasing Sim2Sim gap keeping the offline data collection policy the same. We observe that our $\beta$-DICE algorithm comfortably outperforms closest DICE based baselines.

these environments as the simulator environment. We further create a "real" world environment by changing key configurations from each of these environments. For example, we modify the transition probability in taxi, gravity in cartpole, and link lengths in reacher. These kinds of configuration changes help us assess the performance limits of our algorithm across a variety of sim2sim gap. In table 2, we list all the different sim2sim environments configurations over which we experimented our algorithm. Typically these configurations are such that the real-world performance predicted by the simulator alone is off by 9-45%

We first collect our offline data by using a noisy pre-trained policy which is parameterised by $\delta$. Higher the $\delta$, noisier the data-collection policy. For Taxi and Cartpole environment, we using a uniform random policy for the noise, while we choose zero-mean gaussian policy for continuous environments like reacher and halfcheetah. Using this offline data as well as our simulator, we now evaluate the performance of any target policy, which we parameterise by $\alpha$. Target policy is further defined by a mixture of another pre-trained policy with noise. More the $\alpha$, more the randomness in the policy. Detailed experimental details along with the setup has been detailed in Appendix 9.11

**Results and Observations:** We present the detailed results for all the four environments in figures 3 (Taxi), figures 2a, 4 (Cartpole), figures 2b and 6 (Reacher) and figure 7 (HalfCheetah). For the boxplot, we fix target policy ($\alpha$) and demonstrate the evaluation error for our algorithm across a range of offline dataset ($\delta$) while keeping Sim2Sim gap fixed. For the line plots, we demonstrate the effectiveness of our algorithm across a changing Sim2Sim gap, while keeping the offline

data ($\delta$) and target policy $\alpha$ fixed. We also compare our algorithm against DICE baselines GradientDICE [36], GenDICE [37], DualDICE [35]. DICE baselines are currently the state-of-the-art algorithm in off-policy evaluation and are known to outperform even hybrid off-policy evaluation algorithms. In figure 5, we also compare our algorithm against hybrid off-policy evaluation baselines for the cartpole environment (further details in section 9.11). We observe that our method is able to comfortably outperform closest DICE baselines with the help of extra simulator. These results not only empirically validate the effectiveness of our algorithm, but also point out that we can learn important information from imperfect simulated environments to help in improving RL policies. We also observe that as the Sim2Sim gap increases the performance of our algorithm tends to decline. This means that with increasing Sim2Sim gap the amount of relevant information that can be learned from the simulator diminishes. We observe that this decline actually becomes significant when the Sim2Sim gap breaches the 60% threshold.

### 5.2 Real-world performance validation on Kinova Robotic Arm

**Experimental Setting:** We demonstrate the effectiveness of our $\beta$-DICE algorithm for a sim2real validation task on a Kinova robotic arm. We first collect offline data by asking users to move the arm from one-position to another via RC controllers. Our data collection ensures sufficient coverage of the robotic arm's task space. We then use this offline data along with our in-house gazebo based simulator to experimentally validate the real-world performance of a PID controller using $\beta$-DICE that moves our robot from a given initial location to any desired location.

**Results and Observations:** Our results along with different baselines are averaged over 10 different locations are tabulated in Table 1. We observe that $\beta$-DICE is able to outperform state-of-the-art showing an improvement of 60% over the nearest baseline. There are two key conclusions from all of our experiments. One, although $\beta$-DICE outperforms state-of-the art baselines in off-policy evaluation. We observe that the performance drops when the gap between the target policy and behavior policy increases. Two, prediction error decreases as the gap between the training and test environments increases, as the transferable information between the two environments decreases.

## 6 Limitations

We present the limitations of our work that we wish to address in future work. (1) Our algorithm expects high quality data with sufficient coverage of the state-action space. Identifying the confidence interval of our estimator $w$ will not only ensure sample efficient evaluation, but also help us in designing robust offline reinforcement learning algorithm. (2) Similar to DICE class of min-max optimization, our algorithm also suffers from high variance in their performance. Efforts are required to reduce this variance.

## 7 Conclusion and Future Work

We derive a novel MIS estimator that is able to evaluate real world performance of a robot using offline data and an imperfect robot simulator. We then develop sample complexity bounds, and empirically validate our approach on diverse Sim2Sim environments and Sim2Real environment like KinovaGen3 robot. For future work, we wish to utilize this framework of off-environment evaluation to learn optimal robot policies using simulation and a limited amount of real-world offline data.

## 8 Acknowledgements

The authors thank Neeloy Chakroborty and Shuijing Liu for their valuable suggestions on this paper draft. This work was supported in part by ZJU-UIUC Joint Research Center Project No. DREMES 202003, funded by Zhejiang University. Additionally, Nan Jiang would also like to acknowledge funding support from NSF IIS-2112471 and NSF CAREER IIS-2141781.

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
