# OpenReview forum: "Marginalized Importance Sampling for Off-Environment Policy Evaluation"
_robot-learning.org/CoRL/2023/Conference — CoRL 2023 Poster_

### Official Review · Reviewer_VF29 · 2023-07-13

**Confidence:** 2
**Originality:** Very Good
**Technical Quality:** Excellent
**Clarity Of Presentation:** Good
**Impact:** 3

**Recommendation:**

Weak Accept: I recommend accepting the paper, but will not argue for my recommendation if the majority of other reviewers have a different opinion.

**Review:**

This paper makes a strong technical contribution in the form of the advancement of the MIS methodology of offline policy evaluation in robotics. It has a theoretical component which gives the merits of the algorithm in terms of sample complexity guarantees. It also has extensive experiments on Sim2Sim with ablations for data-collection, target policies and environments, as well as real world Sim2Real experiments on a robotic arm.

My main struggle with this paper were that the mathematical sections were extremely dense, and I found many of the results difficult to follow. For example, Theorem 5.1 is huge, but little description is given to the intuitive significance of each component other than terse definitions of each symbol. I think this paper would be vastly improved by small additions of english-language explanations of each step, and perhaps even mini-summaries briefly, in laymans terms, of what these results actually mean for performance. Section 6 is similar, launching into an optimization derivation "exploiting Fenchel Duality" without giving context as to what we are doing here.

The related work section does a good job placing the work in context of recent MIS work. However, at around 15-lines, I found it to be quite terse and limited in scope. It briefly mentions that model-based methods also exist, but that's about it. I wonder if it could be placed in the broader context of other approaches to long-horizon / high-variance sampling problems. For example, Defensive/Multiple Importance Sampling, & Stratified / Multi-Level Adaptive Splitting?

Similarly, I found the Experiments section to be quite short. The authors have clearly done extensive experimentation, both in the main paper and supplementary material, but I think the paper could be improved by a more detailed analysis/explanation of the results. For example, their main observation is that their method "is comfortably able to outperform...baselines across a range of target policies", and that "as the sim2sim gap increases, the performance drops significantly". Could you expand on the "why"? Additionally, with the line graphs in Figure 2, it actually looks like the baselines may be trending downwards, while Beta-DICE is trending upwards in terms of error. Also for the "length" graph, there is a massive drop in the "simulator" line at 0.0 before recovery. What do these anomalies mean?

### Minor Issues

- Line 248: "Prediction error decreases" -> "increases"?
- 262: "Framwork" -> "Framework"

**Quality Of The Limitations Section:**

Limitations are addressed clearly

**Questions For Rebuttal:**

Mainly covered in the review section: Would appreciate more simplified, english-language explanation of the steps in the sample-complexity proofs, and perhaps more discussion of the results of the experiments.

**Robotics Focus:**

Sufficient demonstration on hardware

**Summary Of Paper:**

Offline Policy Evaluation deals with using offline data collected using some (potentially unknown) behavior policy to estimate performance of a different target policy via importance re-weighting. However, even with recent advances in Marginalized Importance Sampling, such methods often produce extremely large density ratios (which makes learning unstable/hard), and work via indirect supervision (which means they perform poorly when offline data is scarce).

This paper addresses these two issues with the MIS process by proposing a method which breaks the calculation into two factors: Target Policy Occupancy (which has direct supervision) and a second term (which has small magnitude).

The authors provide theoretical sample complexity guarantees, as well as a sequence of experiments on both Sim2Sim (Cheeta, Cartpole etc.,) and Sim2Real (7DOF Kinova Arm) examples to demonstrate the efficacy of their approach empirically.

**Summary Of Recommendation:**

A fairly strong paper with good theoretical backing and extensive experiments. I think presentation could be significantly improved by more careful narrative and explanation in the more mathematical parts, and perhaps some additional analysis to help understand the significance of the experiments.

---

### Official Review · Reviewer_1Zrs · 2023-07-14

**Confidence:** 4
**Originality:** Good
**Technical Quality:** Good
**Clarity Of Presentation:** Fair
**Impact:** 4

**Recommendation:**

Weak Accept: I recommend accepting the paper, but will not argue for my recommendation if the majority of other reviewers have a different opinion.

**Review:**

Strengths:
1. The paper proposes an interesting setting where by they use the ability to generate lots of data in simulator to aid in off-policy evaluation **given that simulator is close to real world**. This is a relevant problem to robotics community with implications in safety, reliability and guaranteeing policy performance for deployed robots.
2. The novelty of the work is in combining two different methods and using their strengths to estimate the distribution ratio: a. Ratio between $\pi_b$ in real world $P_{real}$ with $\pi$ in simulator $P_{sim}$ can be calculated by sampling method for likelihood estimation as we potentially have access to large number of samples for $\pi$ in simulator $P_{sim}$. b. ratio between $\pi$ in simulator $P_{sim}$ and $\pi$ in real world $P_{real}$ is potentially easier to estimate because simulator is close to real world.
3. The authors derive theoretical results on their estimator that gives the error bound of estimate that depend on the distribution shift in simulator and real world.
4. It is shown through simulated and real world experiments that the method outperforms prior method in terms of mean squared error of performance predicted for target policy in real world.

Weaknesses:
1. Improving paper writing: Paper was not easy to read and the notations are inconsistent and confusing.
a. P,P',P_tr, P_te : It is hard to see why there are so many different notations for dynamics when we are only dealing with 2 different dynamics. This made it very hard to read and understand what is going on in the main paper and proof for Theorem 5.1.
b. Simple things like denoting P_tr to P_sim and P_te to P_real can make the paper easier to read.

2. Missing details in Experiments
a. How are the experiments set up? What is the dynamics parameter changed to do the experiments? -- These information are key and missing from the main paper.
b. What is $\beta$-DICE, over which algorithm is the modification for estimating $\beta$ made? How are the baselines used here since the baselines cannot leverage simulation information ?

3. Sparse literature review: There has been a lot of work in this domain and some of which might be relevant baselines and seem to be missing in the paper:
a. DARC: https://arxiv.org/abs/2006.13916
b. https://ojs.aaai.org/index.php/AAAI/article/view/26305
c. https://arxiv.org/abs/2302.08560
d. https://arxiv.org/abs/2001.01866


**Quality Of The Limitations Section:**

Limitations are addressed clearly

**Questions For Rebuttal:**

I have included the questions in the review above.

**Robotics Focus:**

Sufficient demonstration on hardware

**Summary Of Paper:**

The paper proposes a method to improve off-policy evaluation from pre-collected data in real world by leveraging access to the simulator. The paper splits the off-policy distribution ratio in real world between $\pi$  and behavior policy $\pi_b$ in real world $P_{real}$ by first estimating ratio of $\pi_b$ in real world $P_{real}$ with $\pi$ in simulator $P_{sim}$ and then estimating ratio between $\pi$ in simulator $P_{sim}$ and $\pi$ in real world $P_{real}$. It is claimed that this two stage procedures uses information from simulator to better do off-policy evaluation and is shown by experiments on 4 simulated tasks and a real robot experiment.

**Summary Of Recommendation:**

I believe the paper communicates very interesting and relevant idea but needs to improve a lot on writing and presentation. It will make the paper much stronger to have more convincing experiments.

Update: Thanks for updating the paper. I believe the paper has interesting ideas that will be relevant to the community and hence I have increased my score.

---

### Official Review · Reviewer_dLGh · 2023-07-17

**Confidence:** 4
**Originality:** Good
**Technical Quality:** Very Good
**Clarity Of Presentation:** Very Good
**Impact:** 3

**Recommendation:**

Weak Accept: I recommend accepting the paper, but will not argue for my recommendation if the majority of other reviewers have a different opinion.

**Review:**

the paper is concerned with an important topic of evaluating real world performance from offline data.  this would allow to train RL policies offline while keeping in mind the real world performance. The paper is clearly written and the algorithm is easy to understand. The overview image is very helpful as well. The results show that the method can accurately predict real world performance. However the experimental evaluation or the presentation of it is a bit unclear. It would be very interesting to know how much the task that was used to evaluate the methods differs from the offline training data collected in real and in sim. It would also be important to get a sense of how much data is needed as well as what kind of task was evaluated, for example how long and how complicated the task is. this would allow to get a sense for how well the method generalises to out of distribution tasks which would be my main question of the paper.
I also think the paper could improve by placing itself within the related literature and pointing out the main contributions and distinctions.

**Quality Of The Limitations Section:**

Additional details required

**Questions For Rebuttal:**

see above

**Robotics Focus:**

Sufficient demonstration on hardware

**Summary Of Paper:**

The paper presents an algorithm to evaluate real world performance without the necessity to run the RL policies in the real world. To do so it uses marginalised importance sampling on trajectory data from a simulator as well as from the real world. The whole approach is offline and the results show that their approach is accurate in predicting the real world performance of a given RL policy in sim2sim as well as sim2real tasks.

**Summary Of Recommendation:**

the paper is clearly written and easy to follow. The presented work is interesting and focuses on an important problem of sim 2 real transfer. Some details are missing to really asses the applicability of this approach, especially how well this approach generalise to data and trajectories that have not been seen during training.

---

### Official Review · Reviewer_iwBL · 2023-07-17

**Confidence:** 3
**Originality:** Fair
**Technical Quality:** Good
**Clarity Of Presentation:** Good
**Impact:** 2

**Recommendation:**

Weak Reject: I recommend rejecting the paper, but will not argue for my recommendation if the majority of other reviewers have a different opinion.

**Review:**

The authors motivate a clever refactoring of marginalized importance sampling using a simulator where a term can be directly measured.

Strengths:
* Clear motivation for their method drawing from the weakness of current MIS
* Mathematical guarantees on sample complexity and clear explanation of their method
* Empirical results are clearly presented with error bars.
* Multiple sim environments, including manipulation and locomotion.

Weakness:
* Weak baselines since only DICE like methods are considered,
* Method introduces the need for a simulator with relatively low sim2real gap.
* Real baseline includes is an "easy" scenario for OPE with access to the full state-action space.

**Quality Of The Limitations Section:**

Limitations are addressed clearly

**Questions For Rebuttal:**

* Have you tried comparisons to beyond DICE related methods like those described in "Benchmarks for Deep Off-Policy Evaluation"?
* The real world result is very weak since the dataset includes sufficient coverage for state-actions, this almost far from the case in any realistic robotics application. It would be more convincing with a more realistic robotics dataset.
* The limitations do not address the need from a simulator introduced from this method. And the need for relatively low sim2real gap for the method to be effective. This can in many robotics cases be a large cost.

**Robotics Focus:**

Relevant but unlikely to deploy to hardware in near future

**Summary Of Paper:**

In this work, the authors propose a new off-policy evaluation technique, the key breakthrough is to factor the importance sampling into two terms. Occupancy of the target policy in a simulator which can be directly accessed and the ratio of occupancy under simulation and real dynamics. Authors also provide a mathematical bound for the error on such an estimate and show that such a refactoring can be applied to existing OPE algorithms like Dual-DICE.
Comparisons to methods without this estimator for OPE are done in a Sim2Sim setting where transition dynamics are varied for the CartPole and Reacher tasks. Comparisons are made for varying amount of data noise and sim2sim gap and most settings find an improvement over baselines. Improvements are also seen with a real robotics arm reaching task, however, performance drops as the gap between target and behavior policies.

**Summary Of Recommendation:**

The new OPE estimator, $\beta$-DICE, introduced improves over existing MIS methods by removing large density ratio terms and the allowing directly measuring occupancy in simulation. This metric shows improvements in the sim and real evaluations presented in this work. The requirement of a decent simulator adds to the cost of using this algorithm and it is unclear if the improvements warrant it. More robust examples for real world robotics where access to state-actions is scare would make the case a lot more convincing and of interest of a wider robotics audience.

---

### Author Response · Authors · 2023-08-09
**Common Rebuttal for all Reviewers**

We sincerely like to thank the reviewers for their detailed comments on our paper titled Marginalized Importance Sampling for Off Environment Policy Evaluation. Our work is motivated by the need to validate and assess the real-world performance of a robot policy before deployment, thus alleviating safety and cost critical concerns. In this paper, we propose a proof-of-concept framework that estimates  the real-world performance of a robot policy given a real-world offline dataset as well as an inaccurate simulator. To that end, we propose a two-step framework to estimate marginalized importance sampling (MIS) ratio that when incorporated into the reward function, allows us to succinctly calculate the real-world policy performance using just the simulator. We further demonstrate convergence properties of our algorithm by showing the upper-bound on the evaluation error decreases by the order of n^{-1/2}. We also show that our algorithm is flexible–in the sense that it can be combined with any off-the-shelf OPE algorithm and generalizable–in the sense that it gives low evaluation error across a wide-variety of target-policies, offline-dataset as well as sim2real gap (15-60%).

$$ \textbf{Common Concerns} $$

$$ \textbf{Additional Baselines} $$
We compare our algorithm against the state-of-the-art DICE-baselines in off-policy evaluation. DICE baselines use a signature min-max optimization in-order to learn an accurate density ratio to estimate the value of a policy. DICE is known to outperform both classical OPE baselines like importance sampling (IS) methods as well as complex model-based methods.

The reviewers requested that we compare our algorithm against a much-larger hybrid class of algorithms that combine model-based and model-free approaches. In response, we have added two additional baselines: Residual Dynamics and DR-DICE (details below). The results on the Cartpole environment are available here: https://anonymous.4open.science/r/corl2023_rebuttal_136-5213/README.md

- Residual Dynamics: We fit a model from test-environment data by using the simulator as the “base” prediction and only learning a correction term, and use the learned model for OPE.

- DR-DICE: We additionally compare our algorithm against a doubly-robust (DR) MIS estimator [1] that can organically blend the model information with the test-environment data


[1] Ziyang Tang, Yihao Feng, Lihong Li, Dengyong Zhou, Qiang Liu. Doubly robust bias reduction in in-finite horizon off-policy estimation
We can observe that both these baselines DR-DICE as well as Residual Dynamics are comparable or worse than the original baselines.

We can observe that both these baselines DR-DICE as well as Residual Dynamics are comparable or worse than the original baselines.

$$ \textbf{Experimental Setup} $$

We present both Sim2Sim as well as Sim2real experiments in this paper. In the Sim2Sim experiments, we aim to show the effectiveness of our approach across different target policies, offline dataset as well as changing sim2sim gap. We further show the effectiveness of our approach over different types of Sim2Sim environments like Taxi (Tabular), Cartpole (discrete-control), Reacher (continuous control), and HalfCheetah (continuous control) environments. For each of these environments, we refer to the default configurations of these environments as the simulator. We further create a “real” world environment by changing key configurations from each of these environments. For example, transition probability in taxi, gravity in cartpole, link lengths in reacher, etc. These kinds of configuration changes help us assess the performance limits of our algorithm across a variety of sim2sim gap.

| Environment      | Sim2Sim gap Range|
| ----------- | ----------- |
| Taxi      | 15-37%       |
| Cartpole   | 14-60%        |
| Reacher   | 9-86%        |
| HalfCheetah   | 40-45% |


We further evaluate our algorithm over a range of target policies as well behavior policies Behavior policy or data collection policy are the ones that were used to collect offline data. We further parameterise the quality of this behavior policy by $\delta$, which depicts the amount of noise injected into our behavior policy. Similarly, our target policy is the combination of an optimal policy $\alpha$ injected environment specific noise. Here, as before ɑ depicts the amount of noise in our target policy.

---

### Decision · Program_Chairs · 2023-08-30

**Decision:**

Accept (Poster)

**Comment:**

This paper addresses the problem of off-policy policy evaluation, namely evaluating a policy without running it on the real environment, but based on offline data collected from some other behavior policy in the environment. Most existing techniques use some variant of importance sampling to evaluate the policy based on offline data, but this causes estimation error if the offline transitions are too far from the policy's transitions. To address some of these estimation issues with importance sampling resulting partially from distribution mismatch, the paper assumes access to a simulator with low sim-to-real gap, to try to limit this mismatch. Using samples from the simulator has another advantage: it allows computing the importance sampling factor via simulator samples. The paper provides a sample complexity guarantee by upper bounding the evaluation error by 1/sqrt(n), where n is the number of samples in the offline dataset, and also the number of simulator evaluations.

Reviewers were concerned about the breadth of environments and simulated robots this method was evaluated on. Most of the examples in the evaluation section are sim-to-sim with controllable noise levels. The paper has only one scenario of offline data from a real robot, namely a Kinova arm. The authors performed additional experiments to mitigate some of these concerns.

Looking at previous papers in the off-policy evaluation literature, my impression is that a limited number of sim-to-sim scenarios is the norm. While I agree with the concerns that more real data would have made the conclusions of the paper more reliable and robust, the idea of the paper is interesting and relevant for robotics, especially in cases where the sim-to-real gap is small.

I think it is worth accepting this paper as poster.